# Plain Radiography: A Unique Component of Spinal Assessment and Predictive Health

**DOI:** 10.3390/healthcare12060633

**Published:** 2024-03-12

**Authors:** Philip A. Arnone, Andrew E. McCanse, Derek S. Farmen, Mark V. Alano, Nicholas J. Weber, Shawn P. Thomas, Austin H. Webster

**Affiliations:** 1The Balanced Body Center, Matthews, NC 28105, USA; derek.farmen@gmail.com; 2Precision Chiropractic, Williston, VT 05495, USA; drandrew@precisionchirovt.com; 3Coastal Chiropractic Group, Bristol, RI 02809, USA; dr.alano@coastalchiropracticgroup.com; 4NoVa Spine & Wellness, Fairfax, VA 22030, USA; drnick@novasandw.com; 5Long Lake Chiropractic Centre, Nanaimo, BC V9T 2A1, Canada; longlakechiro@shaw.ca; 6Dynamic Life Chiropractic, Royal Oak, MI 48073, USA; awebster@dlchiro.com

**Keywords:** sagittal spinal balance, coronal spinal balance, spinal biomechanical assessment, plain radiographic utilization, postural assessment, spinal assessment, spinal radiographic parameters

## Abstract

While some research supports utilizing plain radiography for measuring biomechanical alignment of the spine for prognosis and treatment, there are contrasting viewpoints regarding both the value and utilization of these procedures in conservative care. Evaluation of both conservative and non-conservative approaches to spinal care revealed vast differences in radiographic utilization and interpretation between orthopedic surgeons, primary care physicians, chiropractic physicians, and physical therapists, which may account for the different viewpoints and rationales in the literature. A research summary is provided to explore any unique biomechanical parameters identified with plain radiography of the spine (PROTS) and how these measurements may relate to patient health. Understanding any unique value provided through biomechanical assessment utilizing PROTS may help chiropractic physicians determine the appropriate use of radiographic procedures in clinical practice and how to coordinate efforts with other conservative and non-conservative spinal healthcare professions to improve patient health.

## 1. Introduction

Altered sagittal and/or coronal balance of the spine (often referred to as spinal deformity) has been shown to cause biomechanical dysfunction, which may be an important factor for spinal health and longevity [1,2,3,4,5,6,7,8]. Altered spinal balance can increase stress to the spine in the form of higher mechanical load and dysfunctional movement patterns, which could contribute to an increased risk of pain, decreased quality of life (QOL), and spinal degeneration [1,3,6,7,9,10,11,12,13]. While there are many approaches to assessing the spine, plain radiography of the spine (PROTS) has long been considered an acceptable diagnostic tool within both conservative and surgical management of musculoskeletal disorders [14].

The global economic impact of musculoskeletal pain is a growing crisis gaining international attention, causing disability in one out of every two individuals affected [15,16]. Neck and lower back pain are the most common musculoskeletal conditions and have been the leading causes of years lost to disability in the world over the past several decades [17,18,19,20,21,22]. Interestingly, the literature suggests that 85% of all chronic lower back pain cases are diagnosed as “non-specific low back pain”, not as a result of injury, but as a result of an unknown cause, typically from spinal biomechanical dysfunction [23]. Given the impact mechanical pain and end-stage spinal degeneration have on world health, measurable diagnostic and prevention strategies are essential in conservative care.

Research related to pelvic parameters has significantly increased over the last 30 years, indicating how relevant these parameters are in spinal health [24]. While there is established value in utilizing PROTS in orthopedic surgery [25,26,27,28,29,30,31,32,33], there are varied opinions regarding the utilization and appropriateness of PROTS in clinical practice within the conservative spinal care professions [34,35,36,37,38,39,40]. Some literature studies suggest that in conservative settings, such as primary care physicians, emergency room services, and chiropractic care, PROTS should be limited to cases in which red flags are suspected, as PROTS may not have any positive impact on acute care outcomes (improved functioning, severity of pain, overall health status, or clinically important cervical spine injury, defined as fracture, dislocation, or ligamentous instability demonstrated with diagnostic imaging) [25,26,27,28,29]. Other studies suggest that PROTS may have value in conservative spinal care beyond screening for trauma and red flags in the form of biomechanical assessment [30,31,32,37].

This review aims to examine any unique biomechanical assessment properties of PROTS, identify the relationship of those values with patient health, and investigate if further research is warranted to understand how these measurements could help advance conservative spinal care. 

Understanding the values of PROTS requires a review of the current literature and utilization of PROTS in spinal healthcare. Since PROTS is utilized in different manners based on the type of health care professional, the various applications of PROTS are evaluated respective to orthopedic surgeons, primary care physicians, emergency room services, chiropractic physicians, and physical therapists.

## 2. Data Collection

Methods of spinal assessment were reviewed using literature studies from PubMed, Index to Chiropractic Literature, and Chiropractic Biophysics (CBPNonprofit.com) with no publication year exclusions. Only papers available in English were included. The initial searches were completed through July 2023. The authors were interested in the following primary outcomes: (1) quantitative results concerning radiographic evaluation of the spine; (2) method and quality of radiographic measurement; (3) method and quality of non-radiographic measurement. Any published literature studies involving spinal radiographic measurements and non-radiographic postural assessment in relation to normal parameters, spinal conditions, and treatment outcomes (including but not limited to improved Oswestry Disability Index (ODI), Health-Related Quality of Life (HRQOL), Visual Analog Scale (VAS), improved function, reduced pain, and disability) were included in the study. Studies were excluded if they did not contain an accurate description of radiographic evaluation or postural assessment or if they were determined to be redundant or unrelated to spinal assessment.

The search strategy was based on concepts that describe radiographic parameters in relation to normal parameters, asymptomatic patients, symptomatic patients, degenerative changes, and surgical outcomes. For example, studies on normal spinopelvic anatomy were located using terms such as “normal”, “healthy”, or “asymptomatic”, while terms for deformities included “kyphosis”, “cervical lordosis (CL)”, “thoracic kyphosis (CK)”, “cervical vertical axis (CVA)”, “lordosis”, or “scoliosis”. In addition, terms used to describe spinal radiographic parameters included “pelvic incidence (PI)”, “pelvic tilt (PT)”, “sacral slope (SS)”, “T1 Slope (T1S)”, “sagittal vertical axis (SVA)”, or “parameters” combined with “pre-operative”, “post-operative”, “chiropractic”, and/or “postural correction”. Additional searches were performed based on references in the reviewed literature to identify studies potentially eligible for our review. Summary reports of selected studies were divided up evenly between the authors. Disagreements regarding inclusion were resolved via discussion.

The authors independently sourced articles related to the search criteria. A shared database was created for the authors to collect, import, and review information. Upon discovery of related articles, the authors independently imported the article title, citation, and summary into the database. The full article was then attached as a PDF to the database for review. A total of 488 articles were imported into the database.

The article database was divided equally among the authors for review for inclusion and exclusion. Each article was summarized independently by the respective reviewers in 1–2 paragraph summaries according to the study objectives. These included the following categories: orthopedic/neurology radiology, radiographic method of the spinopelvic parameters assessment, the quantitative characteristics of the spinopelvic parameters (LL, SS, PI, and PT), quantitative characteristics of cervical–thoracic parameters (T1S, TIA, CL, TK, and CVA), primary care, postural assessment, computerized assessment, physical therapy, spine-related public health, and physical exam. The information was imported into the database so that all authors would have access to the summaries. Once the summaries were completed and inclusion/exclusion parameters were established, the authors collaborated to determine the final inclusion documents. Ultimately, 175 articles were included, and 313 articles were excluded (Figure 1). Included articles were initially reviewed in 2022 and ranged from 1990 to 2022. During the review process, one paper was added from 2022, one from 2023, and one from 2024. 

## 3. Results 

Of the 488 articles that were reviewed, 175 were included. These are divided into two main categories: radiographic methods and non-radiographic methods. The non-radiographic category is further divided into postural assessment methods, computerized postural assessment, physical therapy methodology, and physical exam. The radiographic methods are further categorized into the following sections: cervical spine assessment, thoracic and pelvic assessment, chiropractic clinical rationale, primary care rationale for radiography, and orthopedic methodology. 

### 3.1. Primary Care Radiographic Utilization

There were three publications reviewed related to PROTS in primary care offices [25,26,41]. The literature suggests PROTS for low back pain without indications of suspected serious underlying conditions does not improve function, severity of pain, or overall health status; therefore, primary care physicians should refrain from routine imaging in patients without these red flag indicators. In one study [26], the most common radiographic findings that were reported to patients were the following: discovertebral degeneration (69%), no abnormality (31%), deformity (39%), congenital abnormalities (17%), posterior arch defects (8%), other discovertebral disease (4%), alignment abnormalities (2%), bone formation (1%), sacroiliac disease (1%). There was one paper that discussed the use of cervical spine X-rays in an emergency care and triage setting [27].

### 3.2. Orthopedic Literature and Radiographic Parameters

The vast majority of the literature on the biomechanical measurements of the PROTS is represented within the orthopedic literature. As a result, a review of the literature is essential to understand any unique value of PROTS. Nineteen articles related to radiographic parameters and surgical outcomes published from 2012 to 2022 were included [42,43,44,45,46,47,48,49,50,51,52,53,54,55,56,57,58]. Five articles did not find improved patient outcomes with surgical correction of spinal balance. In the case of single-level lumbar fusion, Rhee et al. found that focal lumbar lordosis and restoration of sagittal balance for single-level lumbar degenerative spondylolisthesis did not yield clinical improvements (ODI, VAS) [55] (see Figure 2; Appendix A). Sielatycki et al. could not correlate a relationship between measured lordosis and Patient-Reported Outcomes (PROs) [46]. Kato et al. reviewed 178 pre- and post-surgical patients and could not demonstrate any post-surgical association in the improved neck disability index (NDI), Short-form 36 (SF-36), or modified Japanese Orthopedic Association score for myelopathy severity between those with and without spinal deformity [47]. According to Rhee et al., the restoration of focal lumbar lordosis and sagittal balance (for single-level lumbar degenerative spondylolisthesis) did not result in clinical improvements based on VAS and ODI. They also acknowledged the lack of “well-powered” studies on the topic within the current literature [55]. Lee et al. found that cervical sagittal alignment and VAS, NDI, SF-36, and Japanese Orthopeaedic Association (JOA) scores were not clearly related following a cervical laminoplasty [50].

A total of eight articles found correlation between corrected spinal balance and improved patient outcomes [26,27,49,52,54,55,57,58]. Alterations in spinal biomechanical measurements unique to PROTS have been identified as a risk factor for developing spinal degeneration, and these measurements have demonstrated value in the surgical setting [33,42,43,44,59,60]. Surgical correction of these imbalances has been shown to improve long-term outcomes by reducing the risk of adjacent segment disease, decreasing functional disability, and improving QOL of scores [43,45,61,62].

Ochtman et al. concluded “…lower PT (pelvic tilt) was significantly correlated with improved ODI (Oswestry Disability Index) and VAS (visual analog scale) pain in patients with sagittal malalignment caused by lumbar degenerative disorders that were treated with surgical correction of the sagittal balance” [54]. Vialle et al. discussed the importance of correct sagittal balance in surgical correction of spinal deformity in both the short term (referencing the gravity plumb line) and long term for preservation of the adjacent levels of the spine [63]. Radiographic attention to post-surgical lordosis also appears to be related to hip degeneration and may improve long-term clinical signs [48]. Ling et al. state that T1 slope, C7 slope, and cSVA are the most important radiographic parameters to be analyzed that affect surgical outcome, which includes physiological alignment of the cervical spine, HRQOL, and NDI [42]. Aoki et al. suggests that post-surgical PI-LL mismatch appears to impact residual symptoms such as lower back pain (LBP), leg pain, and numbness and that maintaining spinopelvic balance should be emphasized in spinal surgery [51]. Merrill et al. suggested that in addition to PI-LL mismatch, the PT-TK relationship is an important factor in maintaining sagittal balance [60]. Kim et al. found that the C2-7 sagittal vertical axis, sagittal morphotype of the cervical kyphosis, and the cervical lordosis minus T1 slope all correlated with HRQOL improvements [43]. One article found that patients with one or two level lumbar total disc replacements with a segmental range of motion (ROM) >5° identified on flexion–extension X-rays had statistically significant better Oswestry Disability Questionnaire and Stauffer–Coventry scores [56].

The last article was related to post-surgical spinal biometrics and hip degeneration. Kawai et al. found that the PI, SS, and PI-LL were associated with risk of increased hip joint narrowing (following spinal fusion) [48].

### 3.3. Radiographic Assessment within Chiropractic Literature and Practice

A total of 19 articles related to the use of radiography within chiropractic practice were reviewed. Of the 19, 4 articles discussed cervical lordosis and/or anterior head translation (AHT) [64,65,66,67]. Saunders et al. stated that head weighting may prove to be a useful therapeutic tool in addressing FHP, assessed using radiographs and the concurrent loss of the normal cervical lordosis [64]. Harrison et al. concluded that cervical traction and spinal manipulation based on cervical radiographs decreased chronic neck pain with improvement of cervical lordosis through segmental and global cervical alignment, as well as decreased anterior head weight-bearing [65]. Fortner et al. found that cervical extension traction, extension exercise, and spinal manipulation based on cervical radiographs improved global cervical lordosis, decreased degenerative cervical kyphosis, reduced neck pain and disability, and improved overall health [66]. Wickstrom et al. concluded there was relief of cervical radiculopathy resulting from non-surgical correction of forward head posture and cervical kyphosis based on radiograph interpretation [67]. One article investigated the relationship between the cervical spine and occlusal contacts and found that changes in posture and occlusion could be observed after the NUCCA chiropractic procedure based on radiographic interpretation [68]. 

Three articles reviewed thoracic hyperkyphosis (THK) [69,70,71], and two correlated thoracic hypokyphosis with the physiological relationship to lung functions [72,73]. Oakley et al., Miller et al., and Oakley et al. all showed a reduction in thoracic hyperkyphosis utilizing PROTS to help determine appropriate application of clinical protocols. Oakley et al. utilized a combination of posture-specific thoracic extension protocols, including mirror image extension traction and exercises, as well as spinal manipulation [69]. Miller et al. used a multimodal rehabilitation program emphasizing mirror image thoracic extension procedures [70]. Oakley et al. also noted improved pain, disability, QOL measures, and VC in some cases, with an average reduction in thoracic cure by 12° [71]. Betz et al. found that with the combination of mirror image traction procedures, as well as corrective exercise and manipulation as a part of CBP technique protocols, the correction of thoracic hypokyphosis/straight back syndrome was achieved. These were consistent with relief of exertional dyspnea and pain [72]. Mitchell et al. concluded that nonsurgical improvement in thoracic kyphosis in a patient with straight back syndrome is possible and that it may positively influence lung capacity, health, and function following a CBP care program [73]. Two articles reviewed the relationship between LBP and lumbar lordosis/flat back syndrome [74,75]. Harrison et al. found lumbar extension traction increased lumbar lordosis measured via lateral lumbar radiographs in patients with chronic LBP associated with hypolordosis, which is a common factor in LBP [74]. Harris et al. also found that improvement in lumbar lordosis, as well as sacral base angle, pelvic tilt, and sagittal balance, simultaneously reduced pain [75]. Two articles focused on scoliosis specifically, with results showing that all patients had a reduction in curvature concomitant with a reduction in pain levels using mirror imaging exercises, traction, and spinal manipulative therapy [76,77].

Normal values for sagittal balance have been established in the literature and can be considered an important patient outcome. Measurements, analysis, and patient positioning were discussed in two articles and found to be reliable and repeatable [39,78]. Another two suggested that PROTS was irrelevant due to the clinicians’ neglect to establish appropriate rationale for utilization and failure to demonstrate improved pain, function, self-reported recovery, HRQOL, and well-being [34,36]. Lastly, one paper discussed the biomechanical evaluation of posture and alignment, then described the six types of subluxation that satisfy Nelson’s criteria, which currently underpin the basis for routine radiographic examination for biomechanical data related to the diagnosis and treatment of patients in modern chiropractic practice [79].

### 3.4. Radiographic Biomechanical Analysis

The review of radiographic biomechanical analysis included 56 studies (see Table 1: Radiographic biomechanical analysis studies). Regarding specific forms of spinal biometrics, there were a number of categories that were identified for the purpose of this section. Several of these categories were highly represented across multiple disciplines; these measurements appeared to be more global in nature and concentrated in the sagittal plane. The number of articles referencing these topics is as follows: sagittal vertical axis: 34, cervical lordosis/T1 slope/cranio-cervical angle: 23, anterior head translation: 28, absolute rotational angle: 26, thoracic kyphosis: 15, sacral base angle/lumbar lordosis/pelvic incidence: 18, Cobb Method and Gore Methods: 11 (see Table 1, Sagittal categories). Lee et al.’s findings were as follows: “T1 slope was a key factor determining cervical spine sagittal balance. Both spinopelvic balance and TI (thoracic inlet) alignment have a significant influence on cervical spine sagittal balance via T1 slope, but TIA (thoracic inlet angle) had a stronger effect than TK (thoracic kyphosis). An individual with large T1 slope required large CL (cervical lordosis) to preserve physiologic sagittal balance of the cervical spine [80]. Additionally, global and sagittal balance are correlated with quality of life and inferior surgical outcome with the most significant measurements being sagittal C7 plumbline and gravity line effecting ODI [1]. Compared to U.S. norms, the Generational decline in physical component summary (PCS) score was more rapid in symptomatic adult spinal deformity patients with no other reported comorbidities. Specifically, PCS scores for patients with isolated thoracic scoliosis were similar to values reported by individuals with chronic back pain, while patients with lumbar scoliosis combined with severe sagittal malalignment demonstrated worse PCS” [81]. Global measurements in the frontal/coronal (AP) plane were markedly less represented in the collated topics. The number of articles referencing these topics are as follows: frontal vertical axis: 7, idiopathic scoliosis: 6, pseudo-scoliosis: 1, and Cobb angle: 6 (see Table 1: Coronal plane). However, an interesting nexus of sagittal spine balance and frontal/coronal plane balance was found by Ma et al. Their conclusions were as follows: “A basic goal in the treatment of spinal deformity is to achieve proper alignment. To achieve this purpose, the surgeon must pay attention to global spinal balance. The following main points can be concluded from the data of this investigation. Children with AIS (Adolescent Idiopathic Scoliosis) showed signs of increased pelvic tilt and decreased TK. In AIS, coronal balance is correlated to sagittal balance. We believe coronal balance and sagittal balance are equally important for decision-making when dealing with AIS” [8].

Segmental spinal biometrics were also dominated by measurements in the sagittal plane, although to a less significant degree. These topics included relative rotational angle: 13 and George’s line: 1 (see Table 1: Sagittal categories). When conditions were the focus of the article, the conditions were as follows. The number of articles referencing the condition follows the condition: Spondylosis/DJD/DDD/central canal stenosis/myelopathy: 17, spinal-related pain/radiculopathies: 15, headaches: 3, idiopathic scoliosis: 5, pseudo scoliosis: 1, spondylolisthesis: 1, vertigo: 1, and TMJ: 1 (see Table 1: Conditions).

When considering the above delineated conditions, predictive mensuration of sagittal alignment appears to dominate the research, although not exclusively. Xing et al. found that T1S (T1 Slope) and TIA could be considered as a constant morphological parameter in the occurrence and development of cervical disc degeneration in the normal population [82]. Another study demonstrated that alterations in the T1 slope are established as an independent risk factor for degenerative cervical spondylytic myelopathy (DCSM) [5]. The findings were similar for the lumbar spine, as reported by Keorochana et al.: “Changes in sagittal alignment may lead to kinematic changes in the lumbar spine. This may subsequently influence load bearing and the distribution of disc degeneration at each level. Sagittal alignment, disc degeneration, and segmental mobility likely have a reciprocal influence on one another” [9]. Clinical interventions for the conditions listed were as follows. The numbers of articles referencing the topic were as follows: therapeutic exercises: 13, spinal manipulation: 12, corrective spinal traction: 12, and spinal surgery: 6 (Table 1: Treatments). Those that showed clinical and or structural improvements included lordosis or kyphosis improvement: 10, symptomatic improvements: 9, telomere length: 1 (see Table 1: Improved symptoms/quality of life).

**Table 1 healthcare-12-00633-t001:** Radiographic biomechanical analysis studies.

Article	Citation #	Sagittal Plane Methods: Sagittal Vertical Axis/Cobb Method/Gore Method/George’s Line/Absolute Rotational Angle	Sagittal Cervical/Thoracic Kyphosis: Cervical Lordosis/T1 Slope/C7 Slope Spino-Crainio Angle/Anterior Head Translation/Cervical Lordosis Improvement	Sagittal Lumbar/Pelvic Lordosis: Sacral Base Angle/Pelvic Incidence	Coronal Plane: Fontal Vertical Axis/Idiopathic Scoliosis/Pseudo-Scoliosis	Treatments: Spinal Manipulation/Spinal Traction/Therapeutic Exercise	Conditions: Spinal Pain/Radiculopathy/Spondylolisthesis/DJD/DDD/Central Canal Stenosis/Myelopathy	Spinal Surgery	Improved Symptoms/Quality of Life
Region		Cervical, Thoracic, Lumbar, Pelvis	Cervical, Thoracic	Lumbar, Pelvis	Cervical, Thoracic, Lumbar, Pelvis	Cervical, Thoracic, Lumbar, Pelvis	Cervical, Thoracic, Lumbar	Cervical, Thoracic, Lumbar	Cervical, Thoracic, Lumbar
Banno T, Togawa D, et al., (2016)	[83]	Yes		Yes					
Berger RJ, Sultan AA, et al., (2018)	[33]	Yes	Yes		Yes			Yes	
Bess S, Line B, et al., (2016)	[81]	Yes		Yes	Yes				
Chun SW, Lim CY, et al., (2017)	[11]	Yes		Yes			Yes		
Daffin L, Stuelcken MC, et al., (2019)	[84]		Yes						
de Schepper EI, Damen J, et al., (2010)	[85]						Yes		
C, F.; Df, L.; M, M.; De, H. (2017)	[86]	Yes	Yes			Yes			Neck Pain, Lower Back Pain, Telomere Length
Fedorchuk C, Lightstone DF, et al., (2017)	[87]	Yes		Yes		Yes	Yes	Yes	Lower Back Pain
Ferrantelli JR, Harrison DE, et al., (2005)	[88]	Yes	Yes			Yes			Neck Pain, Headaches, Lower Back Pain
Fortner MO, Oakley PA, et al., (2017)	[89]	Yes	Yes			Yes	Yes		Neck Pain, Headaches
Fortner MO, Oakley PA, et al., (2018)	[90]	Yes	Yes			Yes	Yes		Dizziness
Fortner MO, Oakley PA, et al., (2018)	[91]	Yes	Yes			Yes	Yes		Neck Pain, Headaches, Lower Back Pain
Glassman SD, Bridwell K, et al., (2005)	[12]	Yes	Yes						
Harrison DE, Cailliet R, et al., (1999)	[2]	Yes	Yes			Yes			
Harrison DE, Cailliet R, et al., (1999)	[92]						Yes		
Harrison DE, Cailliet R, et al., (1999b)	[93]	Yes			Yes				
Harrison DE, Cailliet R, et al., (2002)	[94]	Yes					Yes		
Henshaw M, Oakley PA, et al., (2018)	[95]				Yes	Yes	Yes		Lower Back Pain
Jaeger JO, Oakley PA, et al., (2018)	[96]				Yes	Yes			TMJ
Kang JH, Park RY, et al., (2012)	[97]	Yes	Yes						
Keorochana G, Taghavi CE, et al., (2011)	[9]	Yes		Yes			Yes		
Moustafa IM, Diab AA, et al., (2018)	[98]	Yes	Yes	Yes			Yes		
Knott PT, Mardjetko SM, et al., (2010)	[99]	Yes	Yes	Yes					
Labelle H, Roussouly P, et al., (2005)	[100]	Yes		Yes			Yes		
Lamartina C, Berjano P (2014)	[101]	Yes	Yes	Yes					
Lee SH, Kim KT, et al., (2012)	[102]	Yes	Yes						
Lee SH, Son ES, et al., (2015)	[80]	Yes	Yes	Yes					
Ling FP, Chevillotte T, et al., (2018)	[42]	Yes	Yes						
Liu S, Lafage R, et al., (2015)	[103]	Yes	Yes				Yes		
Ma Q, Wang L, et al., (2019)	[8]	Yes			Yes				
Mac-Thiong JM, Transfeldt EE, et al., (2009)	[1]				Yes				
Maruyama T, Kitagawa T, et al., (2003)	[104]	Yes			Yes			Yes	
Merrill RK, Kim JS, et al., 2017 Sep;7(6):536–42.	[60]	Yes		Yes				Yes	
Miyakoshi N, Itoi E, et al., (2003)	[3]	Yes	Yes				Yes		
Mohanty C, Massicotte EM, et al., (2015)	[4]	Yes	Yes				Yes	Yes	
Morningstar M. (2002)	[105]	Yes	Yes			Yes			Thoracic Spine Pain
Morningstar MW, (2003)	[106]	Yes	Yes	Yes		Yes	Yes		Thoracic Spine Pain
Moustafa IM, Diab AA, et al., (2016)	[107]	Yes	Yes			Yes	Yes		Cervical Radiculopathy
Moustafa IM, Diab AAM, et al., (2017)	[108]	Yes	Yes			Yes	Yes		
Nicholson KJ, Millhouse PW, et al., (2018)	[109]	Yes	yes				Yes		
Oakley P, Sanchez L, et al., (2021)	[110]	Yes	Yes		Yes				
Okada E, Matsumoto M, et al., (2011)	[111]	Yes		Yes			Yes		
Passias PG, Alas H, et al., (2021)	[10]	Yes	Yes	Yes					
Protopsaltis TS, Lafage R, et al., (2018)	[112]	Yes		Yes					
Raastad J, Reiman M, et al., (2015)	[113]						Yes		
Sadler SG, Spink MJ, et al., (2017)	[114]			Yes					
Silber JS, Lipetz JS, et al., (2004)	[115]	Yes	Yes						
Sun J, Zhao HW, et al., (2018)	[5]	Yes	Yes				Yes		
Troyanovich SJ, Harrison D, et al., (2000)	[116]		Yes		Yes				
Watanabe K, Kawakami N, et al., (2007)	[117]	Yes	Yes		Yes			Yes	
Weng C, Wang J, et al., (2016)	[118]	Yes	Yes						
Xing R, Liu W, et al., (2018)	[82]	Yes	Yes				Yes		
Yang X, Kong Q, et al., (2014)	[13]	Yes		Yes			Yes		
Young WF, (2000)	[119]						Yes	Yes	
Yu M, Silvestre C, et al., (2013)	[120]	Yes	Yes	Yes	Yes				
Yu M, Zhao WK, et al., (2015)	[121]	Yes	Yes				Yes		
**Total Number of Articles (n)**	56	48	34	17	11	12	26	6	9

### 3.5. Non-Radiographic Spinal Assessment

Twenty articles were reviewed on non-radiographic methods to assess spinal alignment. Measurements included but were not limited to forward head posture, tragus wall distance, scoliosis, craniovertebral angle, thoracic kyphosis, natural head position, cervical and lumbar curve, sagittal head tilt, and sagittal shoulder-C7 angle (see Appendix B). Four articles directly related to visual assessment from photography or mobile app [122,123,124,125]. Singla et al. found several methods that were reliable to measure various postures [122]. Bryan et al. found that visual postural assessment had a high inter-rater reliability but had low validity utilizing visual assessment for lumbar lordosis using photographs of clothed subjects [123]. Boland et al., utilizing the PostureScreen Mobile^®^ app, found substantial to almost perfect (ICC ≥ 0.81) inter-rater and intra-rater agreement [124]. Stolinski et al. demonstrated good repeatability and reproducibility and checked validity against the Rippstein plurimeter measurements [125]. 

Seven articles addressed visual assessment of body and spinal posture [126,127,128,129,130,131,132]. The tragus wall distance was found to have high intra-rater reliability, but the measurement of a single patient by multiple raters was not supported [126]. Fedorak et al. concluded that “Intra-rater reliability of the visual assessment of cervical and lumbar lordosis was statistically fair, whereas interrater reliability was poor” [127]. Nam et al. found that the assessment of forward head posture was reliably measured between two physical therapists [128]. Yanagawa et al. found that using the flexi-curve for the assessment of thoracic kyphosis showed the reliability being high for kyphosis height, but the reliability for kyphosis length was less (ICC value of 0.54) [129]. Dunn et al. reported that confirmation X-ray was needed for adult idiopathic scoliosis severity rather than solely relying on screening results [130]. Lundström et al. found that trained observer analysis of natural head position and correct head orientation without X-ray can be reliable and intra-observer reliability is often high [131]. Yip et al. found that the craniovertebral angle in subjects with neck pain was smaller than in normal subjects [132]. Seven articles addressed the validity of various measurement systems for body posture assessment [123,133,134,135,136,137,138].

A literature review from Fortin et al. found that most of the studies showed good intra- and inter-rater reliability for measurements taken directly on the person or from photographs, but the validity of the measurements was not always demonstrated [133]. Goldberg et al. concluded that the Quantec system can be useful for monitoring patients as an alternative to radiography, but for the ascertainment of Cobb angles, the two systems are not measuring the same aspect of the deformity [134]. Fortin et al. found that the correlation between 2D and 3D indices was good to excellent (shoulder, pelvis, trunk list, and thoracic scoliosis), fair to moderate (thoracic kyphosis, lumbar lordosis, and thoracolumbar or lumbar scoliosis), and fair to good with Cobb angles and for the trunk list between 2D and radiograph spinal indices [138]. Tools such as the flexi-curve (also known as a flexi-ruler) [129] have been demonstrated to be invalid in the cervical and lumbar region, as they do not accurately represent the total angle of the curve, its shape, or its magnitude [135,136,137]. In the thoracic spine, these methods are reliable and valid [137]. Two articles were focused on devices to measure curvatures of the spine, a surface topography DIERS formetric 4D [139] and the spinal mouse [140]. One article was specific to sitting posture and its effects on musculoskeletal and pulmonary function [141].

### 3.6. Physical Therapy Spinal Assessment

Eleven articles were assessed identifying methodologies utilized by physical therapists in practice to identify and define spinal health. Two articles focused on how and when physical therapists use PROTS [142,143]. This indicated that when physical therapists do order diagnostic imaging, which includes not only PROTS, but also MRI and CT, they are utilized to make clinical decisions including referral, management, or co-management of the patient [142,143]. Two studies did not use diagnostic imaging. One of these studies investigated the use of an ergonomic training program in conjunction with strengthening and stretching exercises on chronic LBP patients. Posture was analyzed utilizing the Zebris WinSpine Pointer Posture method, which quantifies thoracic kyphosis and lumbar lordosis [144]. One was a mini-review of the literature regarding postural re-education in the treatment of scoliosis [145].

Seven articles did use diagnostic imaging. Four articles investigated the use of treatment protocols based on factors found through radiographic examination, with AHT, SVA, C7P, TK, LL, and SS being of particular importance [108,146,147,148,149]. Of these four articles, three investigated lower back pain treatment protocols utilizing SVA, C7P, LL, and SS as defined through radiograph to quantify treatment outcomes with ODI to validate results. One study investigated the treatment of lumbosacral radiculopathy, with the addition of AHT corrective exercises analyzing LL, ODI scoring, back and leg pain, Modified Schober Test, and latency and amplitude of H-reflex (a diagnostic criteria for lumbosacral radiculopathy [147]. One of the studies investigated rehabilitation of patients with suspected cervicogenic dizziness, loss of cervical lordosis (CL), and AHT as defined via radiographic examination, as well as Head Repositioning Accuracy (HRA) deficiencies as defined via the CROM device [108]. Two of the articles investigated treatment of scoliosis, utilizing radiographic examination to define the magnitude of the scoliosis as measured with the Cobb Angle Method [146,150]. Within the reviewed literature studies above, the radiographic assessment of C7P [149], CVA [146], CL [108], TK [145,146], LL [151], and SS [149] did appear to have clinically significant impacts on decreased pain, ODI, Functional Rating Index, HRA, decreased dizziness, vital capacity (VC), and sagittal lumbar curve.

### 3.7. Non-Radiographic Spinal Evaluation Utilizing Physical Exam

Three articles discussed physical exams. There was one article that looked at effects of passive motion analysis and mobilization on cervical lordosis, ROM, and forward head posture on patients who displayed problems in cervical posture [152]. One study examined patients with cervical radiculopathy using orthopedic and neurological tests such as Valsalva’s maneuver, cervical distraction, Spurling’s, or the upper limb tension test. These values should be interpreted with caution in the absence of any other clinical information due to the lack of primary studies investigating the accuracy of these tests [153]. The last study that looked at physical exams evaluated 15 articles that met their inclusion criteria for manual assessment of cervical spine dysfunction. It found that there were methodological weaknesses in their interpretation of reference standards such as radiography, diagnostic nerve block, and reported pain from the subject, as well as in the representative study population [154].

## 4. Discussion

Plain radiography of the spine (PROTS) may provide a unique value in the assessment of spinal health correlations between radiographic measurements, including but not limited to T1 slope, thoracic inlet angle, sagittal vertical axis, cervical lordosis, pelvic incidence angle, lumbar lordosis, pelvic tilt–thoracic kyphosis mismatch, pelvic incidence–lumbar lordosis mismatch, cervical vertical axis, and sacral slope. These are well documented in the literature as potential predictive indicators of future spinal degeneration, functional disability, and quality of life scores. Altered sagittal spinal alignment can increase stress to the spine in the form of higher mechanical load and dysfunctional movement patterns, which may contribute to an increased risk of pain and degeneration [9]. While non-radiographic postural observation demonstrates value, various parameters cannot be adequately measured without the utilization of plain radiography. Considering current suggested primary care guidelines do not include biomechanical considerations, the recommendation to limit PROTS to trauma and red flags may not be appropriate for all conservative spinal healthcare professionals. The orthopedic approach of utilizing PROTS for biomechanical assessment may be more appropriate for conservative care practitioners, such as those in the chiropractic profession, for early detection of spinal deformity with the goal of measuring, predicting, and improving spinal health. In this review, we examined the various values of biomechanical analysis in PROTS, how PROTS is utilized in various spinal care professions for spinal assessment, and its potential application in the conservative management of spinal health in addition to screening for trauma and red flags.

This review included 175 articles. Three publications were related to PROTS in primary care offices. Collectively, these publications suggest PROTS for low back pain without indications of suspected serious underlying conditions is not clinically valuable; therefore, primary care physicians should refrain from routine imaging in patients without these red flag indicators. This may be partially due to the fact that (a) research suggests radiographs for acute, non-traumatic pain do not change outcomes within the first six weeks [25,26,34] and (b) PROTS does not alter the treatment plan in primary care settings [25,41,155]. There are limitations to these studies; despite the importance of spinal balance, radiographic reporting rarely includes any mention of spinal biomechanical abnormalities, with only 2% of reports including comments on alignment abnormalities [26]. Additionally, no specifics were given with regards to the evaluation, treatment strategies, or education level of the physical therapists and chiropractic doctors involved in patient care in the above studies. Therefore, without comparing manual therapy approaches utilizing PROTS vs. not utilizing PROTS, no conclusion can be made on the value of PROTS in the management of acute care patients.

One article discussed the use of cervical spine X-rays in an emergency care and triage setting [27]. Within this study, the Canadian C-Spine Rule was developed as a highly sensitive decision rule for use of C-spine radiography in alert and stable trauma patients in emergency departments (EDs) with 100% sensitivity and 42.5% specificity. The goal of this study was to investigate ways to reduce practice variation and inefficiency in ED use of C-spine radiography. Considering that the primary focus in EDs is immediate lifesaving or life-preserving intervention, this C-spine rule for important C-spine injuries (fracture, ligamentous instability) in the EDs is appropriate. Nevertheless, the C-spine rule neglects postural and biomechanical alterations that may result in mechanical spine pain and/or degenerative changes, and therefore may not be appropriate in conservative care settings that manage mechanical spine problems and long-term spinal health. 

Fifty-six studies related to radiographic biometrics were included in this study. The number of studies on sagittal biometric values is as follows (some studies included multiple biometric values): sagittal vertical axis (n = 34), cervical lordosis/T1 slope/cranio-cervical angle (n = 23), anterior head translation (n = 28), absolute rotational angle (n = 26), thoracic kyphosis (n = 15), sacral base angle/lumbar lordosis/pelvic incidence (n = 18), Cobb Method and Gore Method (n = 11) (see Table 1). Collectively, it appears sagittal biometrics measured with PROTS may have some predictive value. Sheikh et al. stated “Significant associations were found between satisfaction and disability and global coronal and sagittal (sagittal vertical axis [SVA]) alignment” [156]. They later concluded that 2 years post-op, the achievement of global coronal and sagittal alignment was an independent predictor of both satisfaction and disability. This suggests that the early correction of SVA prior to surgery may improve post-surgical outcomes, but more research is needed to confirm this.

Specifically for the cervical spine, the T1 slope relates directly to cervical sagittal balance, as an individual with a large T1 slope requires large cervical lordosis to preserve physiologic sagittal balance of the cervical spine [80,118,157]. Altered T1 slope has been shown to be a predisposing factor in degenerative cervical spondylytic myelopathy (DCSM) [158]. A T1 slope less than 18.5° was an independent risk factor for DCSM [5]. The authors also state T1 slope is “the only parameter showing significant correlation with both spinopelvic balance and TI alignment, which means it is an important parameter influencing TI alignment and spinopelvic balance” [5]. Though currently, to the present study’s authors’ knowledge, there are no studies displaying the ability to non-surgically influence T1 slope, the value potentially lies in the encouragement of preventive strategies—such as exercise and postural awareness—to reduce the risk of DCSM in patients with T1 slopes less than 18.5°.

Additional evidence has shown sagittal balance involves numerous parameters, including pelvic tilt, thoracic kyphosis, and cervical lordosis, with various compensations seen throughout the spine and pelvis when these measurements deviate from normal values [60]. The complex multifactorial mechanisms associated with the development of symptomatic spondylosis (e.g., age, genetics, spinal balance, segmental motion, previous injuries, occupational status, hydration level) create challenges in predicting the rate and effect of development [159]. These articles suggest that biometric values measured in PROTS may have some predictive value for the development of spondylosis, and further studies should be conducted to determine (a) whether changing these parameters reduces the rate of symptomatic spondylosis and (b) if abnormal biometrics, coupled with other risk factors, offer better predictability for developing symptomatic spondylosis. 

Six systematic review articles related to biometrics and post-surgical outcomes published from 2015 to 2022 were included in this review [42,43,44,45,54,55]. Ochtman et al. concluded “…lower PT (pelvic tilt) was significantly correlated with improved ODI (Oswestry Disability Index) and VAS (visual analog scale) pain in patients with sagittal malalignment caused by lumbar degenerative disorders that were treated with surgical correction of the sagittal balance” [54]. However, in the case of single-level lumbar fusion, Rhee and colleagues found focal lumbar lordosis and restoration of sagittal balance for single-level lumbar degenerative spondylolisthesis did not seem to yield clinical improvements [55]. As we see in these examples, the orthopedic literature is not trying to determine whether or not PROTS is valuable, but rather where the value in PROTS is greatest.

Three of the six articles were in relation to cervical sagittal balance and post-surgical outcomes, two of which found that sagittal alignment is associated with quality of life scores, while the third concluded that restoration of cervical lordosis may decrease the incidence of adjacent segment disease [42,43,45]. The sixth and final study discussed the overall importance of spinopelvic alignment considerations, stating, “It is essential to accurately assess and measure these sagittal values to understand their potential role in the disease process, and to promote spinopelvic balance at surgery” [44]. Given that assessment of spinopelvic alignment is important in surgical planning and surgical approach to spinal correction, it is reasonable that greater utilization of these factors be incorporated into conservative spinal care. 

Additionally, outside of traditional standard views, when reviewing patients with one or two level lumbar total disc replacements, it was found that a range of motion (ROM) > 5° identified on flexion–extension X-rays had statistically significant better Oswestry Disability Questionnaire and Stauffer–Coventry scores [56]. As the trend in orthopedic research is to utilize normal values in assessment for surgery, the potential to utilize these measures in non-surgical patients highlights the importance of implementation in conservative spinal care as well. With orthopedics leading the way, more research is needed to investigate PROTS in conservative spinal care for improved segmental motion preservation and long-term patient ODI and Stauffer–Coventry scores.

Twenty articles were reviewed on non-radiographic methods to assess spinal alignment. Measurements included but were not limited to forward head posture, tragus wall distance, scoliosis, craniovertebral angle, thoracic kyphosis, natural head position, cervical and lumbar curve, sagittal head tilt, and sagittal shoulder-C7 angle (see Appendix B). There are mixed results regarding the validity of non-radiographic spinal assessment. Fedorak et al. concluded that “Intra-rater reliability of the visual assessment of cervical and lumbar lordosis was statistically fair, whereas interrater reliability was poor” [127]. Meanwhile, Lundström et al. suggested that trained observer analysis of natural head position and correct head orientation without X-ray can be reliable, and intra-observer reliability is often high [131]. Nam et al. found that there was high inter-rater and intra-rater reliability for visual evaluation of forward head posture and suggests that computer assessment helps increase the value and reliability [128]. 

Computerized and instrumented postural assessments have been shown to have value, though some studies suggest these tools lack specificity when compared to radiographic evaluation. Tools such as the flexi-curve (also called the flexi-ruler) have been demonstrated to be invalid in the cervical and lumbar region, as they do not accurately represent the total angle of the curve, its shape, or its magnitude [135,136,137]. Similarly, posture analysis (specifically with a mobile application), while beneficial as an indicator of postural abnormality, is neither accurate for true measurement of internal spinal angles and measurements nor adequate for spinal correction [124,132]. While useful, non-radiographic postural analysis does not provide accurate, quantitative measures on the cumulative, causative factors that may be contributing to sagittal and global balance, such as PT-TK mismatch, PI-LL mismatch, decreased CL, cervical kyphosis, T1s, TIA, SVA, PT, SS, or PIA, which can only be accurately measured via PROTS. 

Furthermore, other literature studies suggest that PROTS may be able to identify biomechanical imbalances sooner than visual postural assessments in some cases, as the underlying spinal deformity must progress to a higher level of severity before significant postural changes can be observed. For example, anterior head translation (AHT) and a thoracic hyperkyphosis (THK) can be visualized without radiographs and used as predictors for increased spinal pain and headaches [129,152]. However, even in the absence of AHT, loss of cervical lordosis has been associated with altered cervical kinematics, which may contribute to the development of spinal degeneration and reduced HRQOL scores [30,160]. Once cervical kyphosis begins, the deformity tends to perpetuate itself, shifting the head forward and inducing abnormal forces throughout the cervical spine that further progress the deformity [161]. Considering loss of cervical lordosis may increase the progression of cervical spondylosis, early detection and correction of cervical lordosis may reduce risk of future surgical need and improve spinal longevity; however, more research is needed to confirm this correlation. 

Eleven articles were assessed identifying methodologies utilized by physical therapists in practice to identify and define spinal health. Historically, the trend in physical therapy for evaluating spinal balance has not included PROTS, as this is not within the scope of practice of physical therapy [142,143]. However, due to limitations in postural assessment, there has been an increasing number of literature studies in physical therapy that suggest PROTS could be valuable in physical therapy practice [108,142,143,146,147,148,149]. Some literature studies suggested value in PROTS in the form of treatment protocols, with AHT, SVA, C7P, TK, LL, and SS being of particular importance [108,146,147,148,149]. Within the reviewed body of literature, the radiographic assessment of C7P [149], CVA [146], CL [108], TK [145], LL [151], and SS [149] did appear to have clinically significant impacts on decreased pain, ODI, Functional Rating Index (FRI), HRA, decreased dizziness, vital capacity (VC), and sagittal lumbar curve.

Evidence continues to surface that conservative spine care may be able to improve biomechanical imbalances that are identified in postural screening and PROTS [64,68,96,104,145,146,147,150]. For example, correcting abnormal thoracic kyphosis has been achievable, which may negate the negative effects of abnormal sagittal balance of the thoracic spine, such as decreased HRQOL, increased risk of falls, decreased forced expiratory volume in the first second, and complications of osteoporosis [69,70,71,72,73,89]. Oakley et al. published a retrospective case series of 10 patients that showed a reduction in pain levels and disability ratings with an average reduction in hyperkyphosis of 11.3° [69]. 

Conservative practitioners trained in the correction of altered sagittal spinal balance have also shown success in the correction of cervical lordosis and forward head posture, thus improving balance, dizziness, radicular symptoms, headaches, and neck pain in many cases [30,64,65,66,67,86,88,90,91,94,98,105,106,107,108]. Moustafa et al. found that in patients with chronic discogenic lumbosacral radiculopathy, the addition of forward head posture correction to a functional restoration program showed a positive effect on disability, 3-dimensional spinal posture parameters, back/leg pain, and S1 nerve root function [147]. The changes in forward head posture in lumbar radiculopathy management outcomes in experimental and control groups were demonstrably improved radiographically in the intervention group vs. the non-intervention group [147]. Lumbar radiographic parameters have been shown to be correctable as well in conservative treatment, with decreases in lumbar radiculopathy and mechanical back pain and increases in HRQOL scores [74,75,148,149,151]. 

Interestingly, the literature suggests that discovertebral degeneration is 69% of primary findings in radiographic studies [26]. This poses a clinical challenge: there is conflicting evidence regarding the relationship between degenerative changes and spine pain, as imaging findings of spine degeneration are present in high proportions of asymptomatic individuals, increasing in prevalence with age [26,162,163,164,165,166]. Conversely, a 2015 systematic literature review and meta-analysis utilizing MRI demonstrated a greater incidence of symptoms in the presence of disc disease and spondylosis in symptomatic versus asymptomatic adults 50 years of age or younger [166]. Meanwhile, a 2022 study found no association between “age-inappropriate” and “age-appropriate” disc degeneration in terms of LBP [165]. However, this study has two major limitations that affect the strength of its conclusion: (a) the overall prevalence of LBP onset was relatively low, as subjects were drawn from routine health examinations, and (b) the Disc Degeneration Disease (DDD) score utilized in this study was recorded as the sum of all five lumbar levels, which allows for a severely degenerated segment to be rated as mild if combined with four mildly degenerated discs, thus reducing the discriminative power of the LBP diagnosis. 

Regardless of the presence or absence of pain, disc narrowing increases the risk of stenosis and/or radicular compression, zygapophyseal degeneration, motion segment laxity, decreased ROM, and facetogenic pain [167,168,169,170,171,172,173,174]. Furthermore, disc degeneration leads to nerve innervation in growth beyond the outer third of the annulus fibrosis into the inner two thirds [175]. This phenomenon, which increases with degeneration severity, may make the disc more susceptible to nociceptive stimulation, increasing the risk of experiencing discogenic back pain. It is important to note, however, that imaging is only one component of diagnostic testing and needs to be utilized with other diagnostic tests and physical exam findings in order to determine the proper pain generator and appropriate patient management.

The literature suggests that the utilization of PROTS may have unique value in the management of spine health in addition to screening for red flags through the evaluation of spinal parameters. However, there is limited comparative research on postural assessment and radiographic assessment for the correction of altered sagittal and/or coronal spinal balance and any beneficial outcomes that may exist. There are limited high quality studies showing improved long-term health outcomes (including but not limited to VAS, ODI, HRQOL, FRI, HRA, and VC) for patients demonstrating spinal correction as measured utilizing plain radiography compared to patients with “non-corrected” altered sagittal and coronal spinal balance. Additionally, longitudinal health studies also present challenges, as there are many parameters that can affect health long-term. Most conservative research focuses on very few evidence-based techniques that are contributing considerable amounts of research at their own expense.

Although there is research showing that PROTS utilized within the first 6 weeks of conservative care in a primary care physician’s office for acute, non-traumatic spine pain does not improve outcomes, the potential value in PROTS may lie in early detection of altered sagittal and coronal spinal balance, potentially guiding long-term prevention and management strategies [25,155]. Although newer published research is showing a significant genetic component to disc pathology (29–75%), the environmental factors which can be controlled (smoking, ergonomics, BMI, strength, and mechanics) may be of more importance to those genetically susceptible to disc pathology, including spinal balance [159]. Therefore, the primary care guidelines restricting the use of PROTS to only suspected red flags may not be appropriate for conservative practitioners who base care on spinal biomechanics and alignment, such as the physical therapy and chiropractic professions. Future collaborative research between chiropractic, physical therapy, and orthopedic institutes should focus on how conservative practitioners might be able to better utilize PROTS to adequately assess and non-surgically improve spinal alignment, measure the extent of improvement, short-term and long-term outcomes of improvement, and whether early improvement of spinal balance can reduce the risk of developing mechanical spine pain and spondylosis. 

Evidence is provided within this review that there are biomechanical measurements that are unique to the usage of radiographic assessment. The study does have limitations. This paper is not intended to be a full systemic review and as such lacks adherence to these protocols. Given the depth of the published literature on PROTS and the breadth of the topic, this was not an exhaustive review of all the literature available. Only literature studies available in English were included. There is limited research on patient outcomes when utilizing radiographs for biomechanical analysis compared to non-radiographic assessment. As a result, while there is extensive research on the biomechanical parameters unique to radiographs, it is difficult to determine the value this may provide in conservative care. There are also limited studies on the ability of conservative care to demonstrate correction of radiographic parameters. Questions remain surrounding the inter- and intra-practitioner standardization and repeatability of radiographic assessment. Additionally, there are many lifestyle and environmental factors that affect patient outcome that make it difficult to demonstrate improved outcome as a result of correcting the biomechanical alterations recorded with radiographic procedures. The authors felt it pertinent to determine if enough evidence was available to support further research on the utilization of PROTS for biomechanics analysis in conservative spinal care for better patient outcomes. Despite limitations, there appears to be enough evidence that encourages further investigation regarding the utilization and values of radiographic analysis in conservative care. 

## 5. Conclusions

The literature indicates that there are biomechanical measurements that can only be accurately identified with PROTS. Many of these measurements are directly related to patient health and outcome. 

While there is increasing research available demonstrating the ability of chiropractic care and rehabilitative procedures to improve radiographic parameters, there are limited studies comparing the results of spinal care when utilizing radiographic assessment versus non-radiographic assessment. This makes it challenging to determine the full value of PROTS in conservative spinal care. Considering the impact of improper spinal biomechanics on spinal health, future collaboration between orthopedic and conservative practitioners, such as those in the chiropractic profession, could provide benefits from investigating how radiographic biomechanical analysis can be used in order to non-surgically improve spinal biometrics that are associated with spinal health. 

The research associated with the value of PROTS should encourage the chiropractic profession to investigate adopting the orthopedic model of radiographic utilization, which remains focused on measuring and correcting spinal parameters. More research is needed within the conservative spinal care professions regarding the utilization of PROTS for the improvement of spinal biomechanical parameters, as well as its effect on treatment and long-term benefit to patient health outcomes. 

## Figures and Tables

**Figure 1 healthcare-12-00633-f001:**
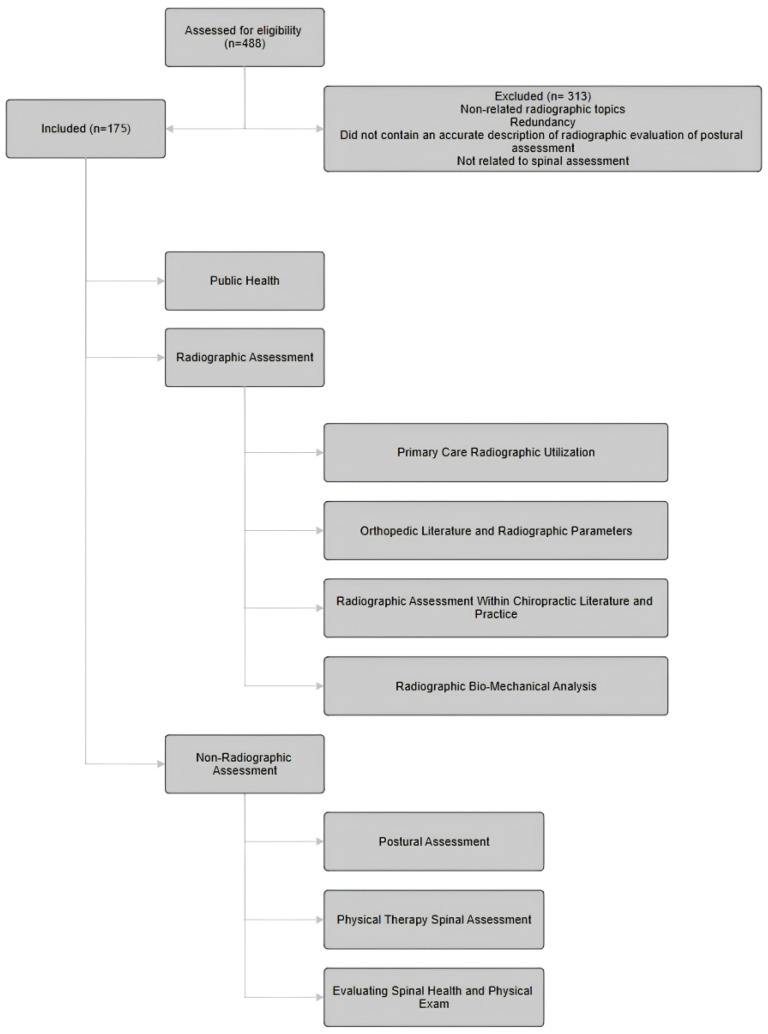
Diagram of data collection.

**Figure 2 healthcare-12-00633-f002:**
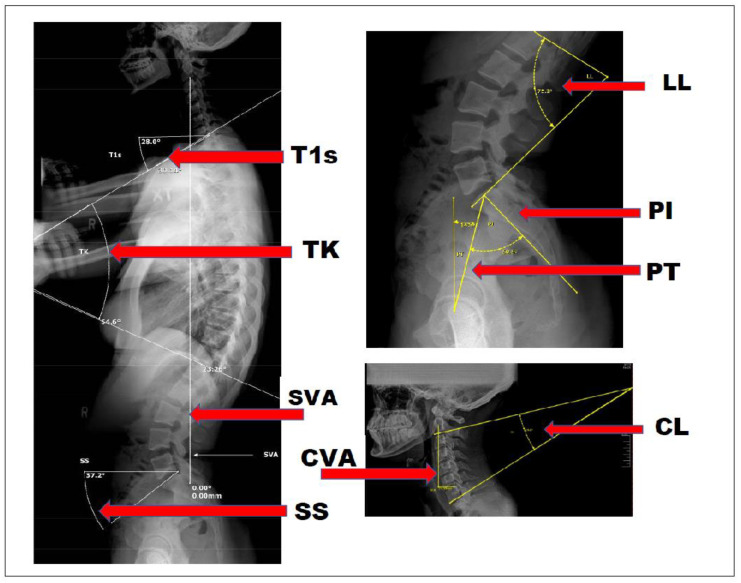
Radiographic measurements. Legend: T1s—T1 slope; TK—thoracic kyphosis; SVA—sagittal vertical axis; CVA—cervical vertical axis; LL—lumbar lordosis; PI—pelvic incidence angle; PT—pelvic tilt; CL—cervical lordosis.

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
