# Peer review of "Plain Radiography: A Unique Component of Spinal Assessment and Predictive Health"

_healthcare, 2024, doi:10.3390/healthcare12060633_

Round 1
Reviewer 1 Report
Comments and Suggestions for Authors
It is well reviewed article.
I have some comments about the conclusion.
References Update: The references need to be updated to include studies from the past five years to ensure the paper reflects the latest research findings.
Advantages and Disadvantages Analysis: It would be beneficial to include an analysis of both the advantages and disadvantages of using Plain Radiography of the Spine (PROTS) in clinical practice.
Patient Group Classification and Treatment Approach: Additionally, there's a need to outline specific criteria for categorizing patient groups that would benefit from additional diagnostic tests. Moreover, discussing how to approach treatment for each categorized patient group in a more practical and clinically relevant manner would enhance the paper's discussion section.
Comments on the Quality of English LanguageNone.
Author Response
Hello Reviewer 1,
Thank you for your review. I have attached a PDF with the responses to your comments.
Respectively submitted
Phil Arnone

Reviewer 2 Report
Comments and Suggestions for Authors
Congratulations on the work, since an extensive review has been carried out. Just a couple of points, in the photograph it would be necessary to put a caption with the legend of the abbreviations (figure 2). Section 4 corresponds better to the title of discussion, since the conclusion is only part of it. It is a very extensive work, so accompanying it with some clinical messages or infographics (as long as the journal's editorial policy allows it) would be very interesting to help the reader make decisions.
Author Response
Hello Reviewer 2,
I have attached a PDF with the responses to your comments. Thank you for your review
Phil Arnone

Reviewer 3 Report
Comments and Suggestions for Authors
The manuscript titled "Plain Radiography: A Unique and Essential Component of Spinal Assessment and Predictive Health" attempts to explore the benefits of plain radiography in chiropractic settings. However, several critical concerns cast a shadow over the manuscript's overall quality and coherence.
Firstly, the review lacks focus, delving into a broad spectrum of subjects without a clear, specific goal. The use of vague and confusing keywords, for example, it was mentioned authors assessed the papers evaluating associations between plain radiography and outcome measures without discussing what measures. This diminishes its validity. The absence of a clear definition for terms like "spinal imbalance" further adds to the confusion, leaving readers uncertain about the authors' intended meaning.
A significant flaw arises from the combination of studies evaluating pre-surgical and post-operative radiography, potentially misleading readers about the context and application of the findings. The absence of a specific strategy for the review exacerbates this issue, undermining the manuscript's overall structure and coherence.
The authors' limited search scope, relying solely on PubMed and chiropractic resources, raises concerns about the comprehensiveness of their research. The lack of information on language selection criteria and the absence of clarity on whether translations were utilized further compromise the precision of the review.
Terms such as "chiropractic radiography" and "diagnostic imaging" are introduced without adequate explanation, leaving readers unfamiliar with these terms confused. The absence of clarification on the types of imaging considered, such as ultrasound or MRI, adds to the ambiguity surrounding the scope of the review.
The title, asserting the high value of plain radiography in diagnosis and treatment, is inconsistent with the findings presented in the manuscript, diminishing its credibility. Additionally, the unrelated conclusion, starting with low back pain, drifts far from the review's initial aim and the articles collected.
Regrettably, these numerous shortcomings collectively render the manuscript unpublishable in its current state. A thorough revision addressing these concerns is essential for the manuscript to meet the standards of scholarly publication.
Comments on the Quality of English LanguageThere are some grammatical errors
Author Response
Hello Reviewer 3,
Thank you for your valuable insight. I have included a PDF to address your comments
Phil Arnone

Round 2
Reviewer 1 Report
Comments and Suggestions for Authors
Great works !
I have no further questions to ask.